# Normal Ovarian Function in Subfertile Mouse with *Amhr2-*Cre-Driven Ablation of *Insr* and *Igf1r*

**DOI:** 10.3390/genes15050616

**Published:** 2024-05-12

**Authors:** Jenna C. Douglas, Nikola Sekulovski, Madison R. Arreola, Yeongseok Oh, Kanako Hayashi, James A. MacLean

**Affiliations:** 1Center for Reproductive Biology, School of Molecular Biosciences, Washington State University, Pullman, WA 99164, USA; jenna.douglas@wsu.edu (J.C.D.);; 2Department of Physiology, Southern Illinois University, Carbondale, IL 62901, USA; 3Department of Cell Biology, Neurobiology and Anatomy, Medical College of Wisconsin, Milwaukee, WI 53226, USA

**Keywords:** insulin receptor, granulosa cells, subfertility, conditional gene knockout, diabetes

## Abstract

Insulin receptor signaling promotes cell differentiation, proliferation, and growth which are essential for oocyte maturation, embryo implantation, endometrial decidualization, and placentation. The dysregulation of insulin signaling in women with metabolic syndromes including diabetes exhibits poor pregnancy outcomes that are poorly understood. We utilized the Cre/LoxP system to target the tissue-specific conditional ablation of insulin receptor (*Insr*) and insulin-like growth factor-1 receptor (*Igf1r*) using an anti-Mullerian hormone receptor 2 (*Amhr2*) Cre-driver which is active in ovarian granulosa and uterine stromal cells. Our long-term goal is to examine insulin-dependent molecular mechanisms that underlie diabetic pregnancy complications, and our conditional knockout models allow for such investigation without confounding effects of ligand identity, source and cross-reactivity, or global metabolic status within dams. Puberty occurred with normal timing in all conditional knockout models. Estrous cycles progressed normally in *Insr^d/d^* females but were briefly stalled in diestrus in *Igf1r^d/d^* and double receptor (DKO) mice. The expression of vital ovulatory genes (*Lhcgr*, *Pgr*, *Ptgs2*) was not significantly different in 12 h post-hCG superovulated ovaries in knockout mice. Antral follicles exhibited an elevated apoptosis of granulosa cells in *Igf1r^d/d^* and DKO mice. However, the distribution of ovarian follicle subtypes and subsequent ovulations was normal in all insulin receptor mutants compared to littermate controls. While ovulation was normal, all knockout lines were subfertile suggesting that the loss of insulin receptor signaling in the uterine stroma elicits implantation and decidualization defects responsible for subfertility in *Amhr2*-Cre-derived insulin receptor mutants.

## 1. Introduction

Insulin receptors are critical activators of several transcriptional pathways and cellular and physiological processes, such as the cell cycle, survival, cell migration, proliferation, and differentiation [1,2]. Insulin receptors (IRs) are tyrosine kinase receptors with four domains. *Insr* and *Igf1r* have high homology in their ligand binding domains, thus allowing alternative ligands, insulin, and insulin-like growth factor-1 to alternatively bind to the receptors [3]. Thus, while INSR and IGF1R have cognate binding partners, our work and others have shown that when one receptor is nonfunctional, the other may effectively bind multiple ligands to preserve function at least partially.

Insulin signaling is essential for reproductive physiology in both males and females. For this work, we will outline the role of insulin signaling in the female reproductive tract, specifically in the ovaries. Similar patterns of hormone cyclicity occur in humans and mice—during the menstrual and estrous cycles, respectively. An essential process in female reproductive biology is the maturation and release of oocytes from the ovary, which is dependent on well-characterized gonadotropin and steroid hormone stimulation. The primary hormones guiding follicle growth and ovulation are follicle-stimulating hormone (FSH), estrogen (E2), progesterone (P4), inhibin, and luteinizing hormone (LH), but the mechanism of insulin and insulin-like hormones is less clear.

When defective, insulin signaling in the female reproductive system can lead to subfertility or infertility due to ovarian and/or uterine dysfunction. Diabetic women have higher risks and rates of reproductive issues, including infertility and pregnancy complications. The Centers for Disease Control report from 2022 indicated that 11% of women in the U.S. 20 years and older have diabetes. Diabetic women have higher risks of ovarian dysfunction, such as enlarged ovaries and potentially reduced ovarian reserve, than women without diabetes [4]. Additionally, there are higher rates of menstruation issues, including amenorrhea, oligomenorrhea, and menorrhagia, in diabetic women [5,6]. Prevalent complications during pregnancy include pre-eclampsia, spontaneous abortion, and neonatal hyperglycemia [7,8]. Offspring from gestational diabetic pregnancies and or hyperglycemia can result in increased risks for metabolic and cardiovascular complications [9]. As type 2 diabetes continues to develop in younger people, the prevalence of women with diabetes in childbearing years is increasing [10]. Obesity is correlated with insulin resistance, elevated insulin, and glucose levels.

Insulin signaling can be dysregulated in cumulus cells in obese and infertile women with polycystic ovarian syndrome without recognizable insulin resistance. The cumulus granulosa–oocyte complex (COC) is altered due to hyperglycemic or hypoinsulinemic conditions. COC defects in type 1 diabetic murine models are characterized by a decrease in oocyte size and meiotic delay, increased apoptotic granulosa cells, an upregulated expression of death signaling proteins, and a decrease in critical gap junction proteins that maintain communication between the oocyte and surrounding nurse cells [11].

Folliculogenesis is promoted by insulin in a stage-specific manner through AKT signaling in vitro [12]. However, studies examining hyperinsulinemia, particularly in women with polycystic ovarian syndrome (PCOS), indicate that granulosa cells become resistant to FSH stimulation. However, when these patients were administered with insulin and pioglitazone treatments for five months, the granulosa cell responsiveness increased [13]. This indicates that granulosa cells are sensitive to insulin resistance, and ovarian function, such as ovulation, is impaired in insulin-resistant environments. Furthermore, in obese women undergoing IVF treatments, hyperinsulinemia in granulosa cells decreased FSH-stimulated functions, such as aromatase activity and a reduced expression of *p-Akt2* [14]. Insulin can sufficiently stimulate E2 and P4 production in granulosa cells [15]. Indeed, even in PCOS patients with peripheral hyperinsulinemia, insulin stimulates E2 and P4 production. These results suggest that insulin signaling and optimal insulin levels are essential for proper granulosa cell function and correlating ovarian functions for cyclicity and maintaining pregnancy.

The Stocco group conditionally ablated *Igf1r* using a combination of *Esr2*-Cre and *Cyp19*-Cre and found female mice were infertile due to a block in antral follicle formation leading to ovulation failure [16]. We subsequently characterized a female reproductive tract conditional ablation of *Insr* and *Igf1r* using progesterone receptor Cre (*Pgr*-Cre) where granulosa cell-specific ablation occurred after the block revealed in the prior study. These mice exhibited subfertility in *Igf1r* mutants and complete infertility in *Insr*/*Igf1r* double mutants (DKO) [17]. The infertility was characterized by a significant reduction in ovulation (50% relative to controls) with follicles exhibiting trapped oocytes in partially luteinized tissue that were not released and ultimately underwent atresia. Further, while the uterus appeared normally receptive, oocytes that were ovulated and fertilized never implanted in DKO, and the fecundity of *Igf1r* mutants was severely reduced [18].

Here, we evaluate fertility and ovarian function in the *Amhr2-Cre-*mediated ablation of *Insr*, *Igf1r*, and double receptor knockout female mice. *Amhr2-Cre-*driven ablation should occur during the secondary to antral follicle transition, placing the timing of insulin receptor dysfunction in between the two prior models. Our findings demonstrate that while these mice are subfertile, the loss of insulin-dependent signaling in granulosa cells during this window elicited by *Amhr2*-Cre is not sufficient to impede ovarian cyclicity, folliculogenesis, and ovulation.

## 2. Materials and Methods

### 2.1. Mice

All mice used in this study were maintained on a C57BL6 genetic background. All animals were housed under a 12:12 light–dark cycle at 70% humidity. Genotyping was conducted by collecting genomic DNA from tails and toes 5–8 days of age (Appendix A), as described previously [17]. Mice with floxed alleles for *Insr* (#006955, [19]) and *Igf1r* (#012251, [20]) were obtained from the Jackson Laboratory. *Esr2*-Cre mice [21] were kindly provided by Jay Ko (University of Illinois) but are now available from the Jackson Laboratory (#028717). *Pgr*-Cre mice [22] were kindly provided by Dr. John Lydon (Baylor College of Medicine). *Amhr2*-Cre mice (014245-UNC, [23]) were obtained from the MMRRC as frozen embryos and rederived by the WSU animal production core.

Puberty was assessed by the timing of vaginal opening. Mice were generated for single and double conditional knockout by breeding *Amhr2-*Cre-floxed males to floxed females. Control females possessed variable genotypes, such as *Insr f/f*, *Igf1r f/+*, *Amhr2 +/+*; *Insr f/+*, *Igf1r f/f*, *Amhr2 +/+*; *Insr f/f*, *Igf1r f/+*, *Amhr2 +/+*; and *Insr f/+*, *Igf1r f/+*, *Amhr2 Cre/+*, all of which exhibit similar traits as stock C57BL6 animals in our current and prior analyses. For simplicity, *Insr f/f*, *Igf1r f/+*, *Amhr2-Cre/+* (*Insr* conditional knockout); *Insr f/+*, *Igf1r f/f*, *Amhr2-Cre/+* (*Igf1r* conditional knockout); and *Insr f/f*, *Igf1r f/f*, *Amhr2 Cre/+* (*Insr/Igf1r* double conditional knockout) are abbreviated as *Insr^d/d^*, *Igf1r^d/d^*, and DKO, respectively.

### 2.2. Estrous Cycle Assessment

Vaginal smears were collected for 30 days beginning seven days post vaginal opening for each genotype. Vaginal canals were flushed three times with 1 × PBS and transferred to a microscope slide at the same time each morning. Cell composition was observed and scored under bright field microscopy as previously described [24]. Estrous stages were classified as follows: proestrus, primarily nucleated and some cornified cells; estrus, primarily cornified epithelial cells; metestrus, cornified epithelial and leukocyte cells; and diestrus, primarily leukocytes.

### 2.3. Superovulation and Ovulation Assessment

Ovulation was induced in mice through injections of equine chorionic gonadotropin [eCG (PMSG), Biovendor RP178272, Asheville, NC, USA] and followed by a single injection of human chorionic gonadotropin (hCG, C0434 Sigma St. Louis, MO, USA), as described previously [17,25]. Briefly, female mice weighing ~15 g (aged 21–28 days) were injected with 5 IU eCG and 48 h later, a single injection of 4 IU hCG. Mice were euthanized and collected at 12 h post HCG injection for molecular and histological studies. For cumulus–oocyte complex (COC) retrieval, oviducts were removed and transferred to a dish containing PBS. The ampulla mechanically burst and was flushed of COC using PBS injected via a 30 G blunt-end needle. COCs were transferred to M2 media (Sigma M7167) at room temperature and imaged. One ovary was snap-frozen in liquid nitrogen and stored at −80 °C for RNA isolation with Trizol (Invitrogen, Waltham, MA, USA) according to the manufacturer’s instructions and the other preserved in 4% paraformaldehyde for embedding and histological analyses, as described previously [17,25].

### 2.4. Follicle Analysis

Ovaries were serially sectioned at 5 µm, and every 5th section was transferred to a microscope slide. Forty sections are obtained per sample. Sections were stained with hematoxylin and eosin and counted as previously described [13]. Images were processed using the Image J: Cell counter plug-in. Primordial follicles were defined as type 1, primary as type 2, secondary as type 3, antral as type 4, and corpus luteum as type 5. Scored image uploads generated by the algorithm were manually verified for consistency by two observers.

### 2.5. Immunohistochemistry—INSR, IGF1R, HSD17B7

Before staining, sections were deparaffinated in xylene and rehydrated in ethanol 100%, 95%, 75%, and 1 × PBS. Antigens were retrieved in pH 6 citrate buffer using an automated decloaking chamber (Biocare Medical, Pacheco, CA, USA) according to the manufacturer’s instructions. Protein localization was performed using commercially available antibodies against INSR (Abcam, Cambridge, UK ab137747, 1:1000), IGF1R (Cell Signaling Technology, Danvers, MA, USA #3027, 1:500), and HSD17B7 (EMD Millipore, Burlington, MA, USA ABS1009, 1:2000), as described previously [17]. Bound antibodies were visualized using Vectastain secondary biotinylated secondary antibodies and visualized through a Vectastain ABC kit (Vector Laboratories, Newark, CA, USA). Negative controls were performed, substituting the primary antibody for preimmune on non-specific rabbit serum, as available. Sections were counterstained with hematoxylin and eosin.

### 2.6. Apoptosis Assay

Apoptotic cells were identified in ovarian sections using the terminal deoxynucleotide transferase dUTP nick end labeling (TUNEL) assay that was performed according to the manufacturer’s instructions using the ApopTag^®^ Fluorescein In Situ Apoptosis Detection Kit (S7110; Millipore, Burlington, MA, USA). Apoptotic cells were counted using the ImageJ (Version 1.54h): Cell counter plug-in. Antral follicles with at least one positive cell were scored as positive, and the total apoptotic cells were counted within each positive follicle.

### 2.7. Quantitative Real-Time RT-PCR (qPCR) Analysis

RNA was extracted from the whole ovarian tissue homogenized in 500 µL TRIZOL. RNA was extracted using Phasemaker^TM^ tubes (Thermo Fisher, Waltham, MA, USA) following the manufacturer’s protocol. cDNA was synthesized from 3 µg of total RNA using the High-Capacity cDNA Reverse transcription kit A (Applied Biosystems, Waltham, MA, USA). Relative mRNA expression was analyzed using the BIO-RAD CFX Opus 96 Real-Time PCR System using Applied Biosystems’ corresponding SYBR Green Master Mix. Relative expression was normalized against *Rpl19* using the 2^ΔΔCT^ method. Gene-specific primers are shown in Appendix A and were designed and used as previously described [17].

### 2.8. Statistical Analysis

All qPCR, histological measurement, and fertility assessment data were subjected to one-way ANOVA Prism 9.0 (GraphPad, San Diego, CA, USA). Comparisons of means between two groups were conducted using *t* tests, and differences between individual means of multiple grouped data were tested by a Tukey multiple-range post-test. All data met the necessary criteria for ANOVA including equal variance as determined by Bartlett’s test. All experimental data are presented as the mean ± SEM. Unless otherwise indicated, a *p* value of less than 0.05 was considered statistically significant.

## 3. Results

### 3.1. Fertility Analysis of Female Mice with Conditional Ablation of Insulin Receptors

To examine the impact of the loss of INSR and IGF1R signaling on female fertility that avoids the severe defects of global receptor deletion [26,27,28] and the spatial and temporal fertility blocks identified by previous conditional knockout studies [16,17,18], we used *Amhr2*-Cre to ablate *Insr* (*Insr*^d/d^) and *Igf1r* (*Igf1r*^d/d^) individually and eliminated the potential for redundancy from receptor cross-activation by generating double receptor knockouts (DKO). We found no differences in mating behavior in receptor mutants relative to controls (Figure 1A). While INSR and IGF1R were significantly ablated specifically in the uterine stroma, no differences in gross uterine morphology or diameter at estrus were observed, and a normal distribution of uterine glands was present. All three mutant lines were capable of generating pups, but there was a significant delay in the timing of birth in DKO mice that held their litters 1.45 ± 0.37 days longer than control mice (Figure 1B). Mean litter sizes were significantly reduced in all three mutant lines with *Insr*^d/d^ producing two fewer pups per litter and *Igf1r*^d/d^ and DKO mice producing three fewer pups per litter (Figure 1C).

The *Amhr2*-Cre driver is currently the best option for conditional knockout studies seeking to examine gene function in stromal cells of the uterus [23]. Recently, this Cre line has controversially been demonstrated to elicit a global deletion of some targets [29,30,31]. In our study, a spurious embryo-wide inactivation of *Insr* and *Igf1r* did not occur as mothers were normal (i.e., not infertile dwarves as may be expected for complete insulin receptor mutants [27,32,33]), and we carefully genotyped mice to ensure the floxed allele existed outside of reproductive tissues. However, active Cre recombinase is present in secondary follicles of adult ovaries [34]. As prior studies revealed ovulatory defects when *Insr* and *Igf1r* were conditionally deleted in granulosa cells of primary follicles and in luteinizing granulosa cells, we sought to determine whether *Amhr2*-Cre-mediated deletion in the ovary could contribute to the observed subfertility.

### 3.2. Evaluation of INSR and IGF1R Receptor Ablation in Ovarian Granulosa Cells

In control animals, nearly all granulosa cells were found to be IGF1R-positive, and 80% were positive for INSR. Some INSR and IGF1R ablation was observed in secondary follicles and was the most prevalent in antral follicles in accordance with where *Amhr2*-Cre activity was expected to have been the most effective (Figure 2A). The levels of receptors were reduced but not fully ablated in granulosa cells of follicles in DKO mice. We did not examine single knockout mice as the efficiency of ablation for each single gene was expected to be similar in DKO mice. INSR was eliminated in 27% of granulosa cells in each antral follicle, and IGF1R was eliminated in 36% of granulosa cells in each antral follicle (Figure 2B). Thus, we hypothesized that most INSR- and IGF1R-dependent processes may be preserved in *Amhr2*-Cre-derived conditional knockouts as the majority still possessed one or both receptors, unlike our previous ablation model using the stronger *Pgr*-Cre where receptor ablation was predominant throughout antral and periovulatory follicles and nearly complete in granulosa cells, and subsequent defects in luteinization and ovulation were observed [17]. To test our hypothesis, we assessed ovarian function in insulin receptor mutants using the most common physiological and molecular assays. We found no evidence of gross ovarian morphology or size. A histological analysis of ovarian sections from all four genotypes showed no obvious signs of developmental block. A subsequent quantitative analysis of follicles in DKO ovaries found no differences in the stage distribution relative to controls (Appendix A).

### 3.3. Analysis of Estrous Cyclicity and Ovulation in Conditional Knockout Females

While *Amhr2*-Cre is not expected to be active and ablate insulin receptors in the pituitary and there was no obvious block in folliculogenesis, we sought to rule out subtle anomalies in ovarian function by charting ovulatory cycles and subsequent ovulation from each genotype. First, we examined the progression of the stages of the estrous cycle and revealed normal hormone cyclicity and progression through estrous cycle stages. Representative patterns from each genotype are shown in Figure 3A. In contrast to mice with *Igf1r* ablation using *Esr2*-Cre where ovulation is inconsistent and stalls in metestrus for extended periods (Figure 3B), all *Amhr2*-Cre-derived mutants exhibited largely normal 3-4 day cycles. This is in agreement with our prior study examining insulin receptor ablation with *Pgr*-Cre (Figure 3C) which acts later and more strongly than Amhr2-Cre but ultimately did not impact estrous cyclicity despite oocytes not being released efficiently prior to luteinization [17]. To further discern ovarian cyclicity, we quantified the time spent in each phase of the estrous cycle. No significant differences existed in the estrus, proestrus, or metestrus stages (Figure 3D). The exception was a mathematically significant prolonged diestrus stage in both *Igf1r*^d/d^ and DKO mice. A similar pause at the diestrus stage was observed in *Pgr*-Cre-derived *Igf1r*^d/d^ and DKO mice in our previous study [17]. As those mice exhibited a 50% reduction in ovulation, we quantified the number of cumulus–oocyte complexes in the oviduct. To rule out an upstream disruption of ovulation and to synchronize the mice, we used exogenous gonadotropins to hyperstimulate ovulation and assess key genes that promote ovulation.

### 3.4. cKO Females Respond to Exogenous Hormone Supplementation

To determine whether the normal estrous cycles we observed indeed correlated to successful ovulation, we superovulated female mice, excised their oviducts, and flushed cumulus–oocyte complexes (COCs) for morphological assessment and quantification. All three conditional knockout lines produced COCs that had intact granulosa cell layers and were indistinguishable from control COCs. There were no significant differences in the number of COCs retrieved from any group (Figure 4A). We never observed oocytes trapped in corpora lutea in *Amhr2*-Cre DKO ovaries, in contrast to the *Pgr*-Cre DKO mice in which they were found frequently [17]. Indeed, the number of COCs retrieved matched very closely with the number of CL present on the ovarian surface with COC/CL ratios of 91 ± 1.5% in control animals and 87 ± 3.2% in DKO mice (*n* = 5). In addition to the absence of trapped oocytes, luteinization appeared more uniform within CL cross-sections as assessed by immunohistochemistry for the luteal cell marker HSD17B7 (Appendix A). In *Pgr*-Cre DKO mice, HSD17B7 staining was sporadic indicating an altered timing of luteinization, and those mice exhibited a reduction in progesterone production and the downregulation of ovulation-promoting factors and several enzymes in the progesterone synthesis pathway [17]. We similarly assessed the ovarian expression of *Pgr*, *Lhcgr*, and *Ptgs2* at 12 h post-hCG in hyperstimulated mice and found no significant difference between control, single receptor, and DKO mice (Figure 4B). A modest but statistically insignificant increase in Star expression was observed in *Igf1r*^d/d^ and DKO mice (Figure 4C). However, all six genes examined in the subsequent steroid hormone synthesis pathway were unaltered in any knockout mice line.

Prior insulin receptor ablation studies using *Esr2*-Cre observed a significant decline in follicle health with widespread apoptosis contributing to the failure of antral follicle development [16]. In our *Pgr*-Cre-mediated knockout study, there was no significant increase in atretic follicles in DKO mice, and granulosa cell apoptosis, while commonly observed, was highly variable such that no follicle type exhibited an increase in TUNEL-positive cells relative to controls [17]. To further determine follicle quality in *Ahmr2*-Cre *Insr* and *Igf1r* conditional knockouts, we counted the number of apoptotic follicles per tissue section in each genotype and found no differences in the prevalence of follicles with TUNEL-positive cells in any genotype. However, when the number of apoptotic cells were counted per positive follicle, there was a ~3-fold increase in the number of positive cells in antral follicles of *Igf1r*^d/d^ and DKO mice (Figure 5). Only a few follicles exhibited an extreme proportion (i.e., one-third) of dying granulosa cells in agreement with overall normal ovarian function and ovulation in all three mutant genotypes.

## 4. Discussion

Folliculogenesis and ovulation are regulated by the functional hormone cyclicity of LH, FSH, E2, and P4. Disruption in this process can occur through metabolic disorders, diabetes (T1D, T2D, GDM), and PCOS. Many studies have shown the relationship between metabolic dysregulation and ovarian defects, such as abnormal menstruation, amenorrhea, oligomenorrhea, and menorrhagia, in diabetic women [4,5,35]. These studies have used diabetic models of mice or human diabetic patients. These studies characterized whole-body insulin dysregulation. However, few studies have investigated ovarian tissue with altered insulin signaling while keeping the peripheral body insulin-dependent pathways functioning normally. In this study, we conditionally deleted *Insr* and *Igf1r* using *Amhr2*-Cre which is active in granulosa cells of the ovary and in uterine stromal cells. *Amhr2*-Cre is presently the best option for eliciting conditional gene ablation in uterine stroma cells, but the potential confounding effects of ovarian deletion on female fertility must be accounted for to completely characterize subfertility and infertility in subsequent mutant mice.

We expected that *Amhr2*-Cre would sufficiently ablate insulin receptors in the ovary eliciting an ovulation defect based on prior studies from our lab and others. The conditional ablation of *Igf1r* using *Cyp19*-Cre and *Esr2*-Cre was performed to test the hypothesis that IGF1R signaling was essential for antral follicles to respond to FSH in vivo [16]. The ablation of *Igf1r* with either Cre driver resulted in significant subfertility and the combination of both infertilities. While FSH receptor expression was not altered, the FSH-dependent control of granulosa proliferation and differentiation was crippled, and follicles did not progress to the antral stage resulting in ovulation failure. Our subsequent work showed that the *Pgr*-Cre ablation of both *Insr* and *Igf1r* was necessary to eliminate the potential masking of ovarian phenotypes due to the cross-reactivity of INS or IGF ligands with their non-cognate receptor when their high affinity partner was absent [17]. In these mice, the later activity of *Pgr*-Cre allowed *Igf1r*^d/d^ mice to skip the block observed by the Stocco group [16,17]. Still, these mice exhibited substantial subfertility in single receptor mutants, and DKO mice were completely infertile. Both of these Cre models were characterized by elevated granulosa cell apoptosis, a reduced expression of ovulation-promoting genes, a reduced expression of steroidogenesis enzymes, and ultimately estradiol and progesterone production were compromised. However, in contrast to the *Esr2*-Cre ablation of insulin receptor signaling, this did not elicit abnormal estrous cycles in *Pgr*-Cre mutant lines, similarly to our findings in this study after *Amhr2*-Cre ablation.

We did find that the loss of INSR and IGF1R in ~one-third of antral follicle granulosa cells did correlate with a 3-fold increase in the incidence of apoptotic cells in each follicle. However, this mosaic deletion and subsequent loss of mural granulosa cells did not appear to impact follicle health significantly as there was no increase in atretic follicle counts in any genotype. Similar follicular distribution and the lack of induced atresia were also features of the stronger but later acting *Pgr*-Cre used in our prior study [17]. We did not assess whether the loss of insulin signaling and granulosa cell death impacted subsequent thecal or luteal cell development. However, the lack of change in any of the steroidogenesis factors and ovulation-promoting genes would indicate that developing follicles progressed normally and could be capable of supporting oocyte maturation to fertilization competency. Follicles reaching the periovulatory phase were found to extrude their oocytes with maximal efficiency in *Amhr2*-Cre DKO as no trapped oocytes were observed within corpora lutea. Like *Pgr*-Cre, *Amhr2*-Cre could potentially ablate INSR and IGF1R in the oviduct. We did not assess this histologically, but there was no evidence that the oviduct was less coiled, shorter, or retained oocytes/embryos in either our prior study or this one.

We anticipated that ovulation might stall in *Amhr2*-Cre DKO mice, as we observed in *Pgr*-Cre DKO mice [17]. However, all lines of evidence presented in this paper suggest that ovulation is largely normal in *Insr*^d/d^, *Igf1r*^d/d^, and DKO mice due to the relatively reduced activity of *Amhr2*-Cre and greater residual expression of INSR and IGF1R in secondary and antral follicles after conditional ablation.

In agreement with prior studies, the disruption of the IGF1R axis has a greater impact on reproductive parameters than INSR alone. However, differences between single mutants and DKO mice indicate that some redundancy is present that should be investigated further. Taken together, our present findings indicate that the *Amhr2*-Cre model is likely useful for examining insulin receptor action in implantation and postimplantation developmental processes, which we are presently pursuing.

## 5. Conclusions

Female mice harboring *Amhr2*-Cre-dependent deletions of *Insr* and *Igf1r* are significantly subfertile. However, estrous cyclicity, ovulation rates, hormone production, and luteinization all appear to be indistinguishable from control mice. Thus, it is likely that the decline in litter size is due to failed implantation or early embryo loss associated with the ablation of INSR and IGF1R in uterine stromal cells where *Amhr2*-Cre is also active.

## Figures and Tables

**Figure 1 genes-15-00616-f001:**
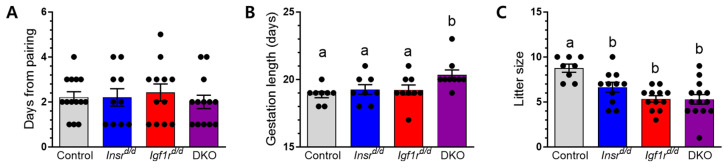
Female insulin receptor mice are subfertile. Control females and those lacking one or both insulin receptors were bred to males of established fertility. (**A**) All female mice presented with post-coitus vaginal plugs to indicate successful mating. Data shown are the mean delay for the appearance of the first plug (*n* = 10–15). (**B**) After mating, the pairs were separated, and the gestational length was measured from plug date to birth across all genotypes (*n* = 8–9). (**C**). The number of pups produced per litter from successful pregnancies (*n* = 7–16). (**A**–**C**). Bar heights indicate the mean ± SEM for each genotype. Letters denote means that are significantly different (*p* < 0.05), one-way ANOVA with a Tukey multiple comparison post-test.

**Figure 2 genes-15-00616-f002:**
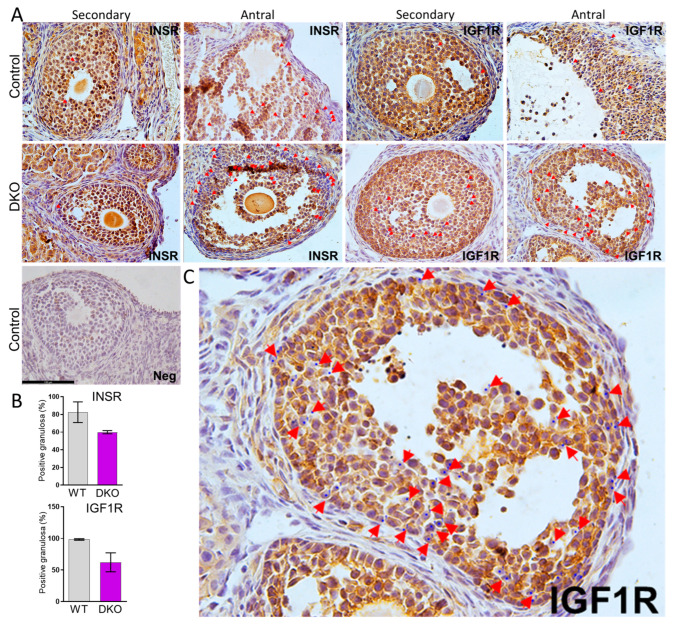
INSR and IGF1R are reduced in granulosa cells of DKO ovaries. (**A**) INSR and IGF1R residual proteins were detected by immunohistochemistry with antibodies specific to each receptor. As differences were not easy to visualize in full ovarian cross-sections, positive and negative granulosa cells were counted in individual secondary and antral follicles in high magnification images (*n* = 4 animals/genotype). Red arrows indicate granulosa cells were receptor expression is absent. (**B**) DKO mice exhibited a decrease in the percentage of INSR- or IGF1R-positive granulosa cells in antral follicles. Data are expressed as the mean ± SEM of the average percentage of positive granulosa cells found in each follicle per animal (at least 10 follicles per animal). (**C**) An enlarged image from panel A is provided to distinguish positive and negative cells.

**Figure 3 genes-15-00616-f003:**
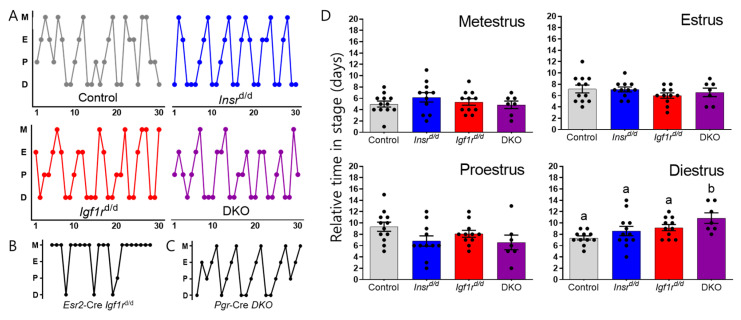
Estrous cycles were monitored for 30 days (*n* = 7–12). (**A**) Representative plots for each genotype are shown indicating the transition from metestrus (M) to estrus (E) to proestrus (P) and to diestrus (D). (**B**) For comparison, the ablation of *Igf1r* using *Esr2*-Cre is shown, which acts earlier than *Amhr2*-Cre in granulosa cells of activated follicles and in the developing ovary, has ovulation failure. (**C**) For comparison, the ablation of both *Insr* and *Igf1r* using *Pgr*-Cre is shown, which acts later than Amhr2-Cre in granulosa cells of activated follicles and in the developing ovary, has partial ovulation impairment. (**D**) The number of days at each stage of the estrous cycle were plotted for each animal. Bar heights indicate the mean ± SEM for each genotype. Letters denote means that are significantly different (*p* < 0.05), one-way ANOVA with a Tukey multiple comparison post-test.

**Figure 4 genes-15-00616-f004:**
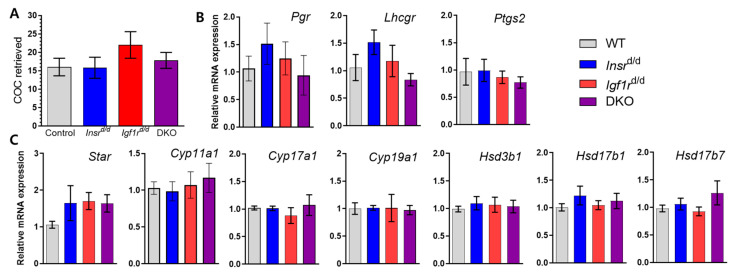
Ovulation is normal in Amhr2-Cre-mediated insulin receptor knockout mice. Mice were superovulated and sacrificed 12 h post-hCG for histological and molecular analyses. (**A**) Oviducts were extirpated, flushed with PBS, and retrieved COCs were counted (*n* = 8 control, *n* = 5 for each conditional knockout). (**B**) qPCR analysis of established ovulation-promoting genes in conditional insulin receptor knockout mice relative to control which was arbitrarily set to 1 (*n* = 10–12 per genotype). (**C**) qPCR analysis of rate-limiting enzyme for steroid synthesis, Star, and steroid hormone synthesis pathway genes relative to control which was arbitrarily set to 1 (*n* = 12 per genotype). (**A**–**C**) Bar heights indicate mean ± SEM for each genotype with no significant differences (*p* > 0.05) observed between control and mutants determined by one-way ANOVA with Tukey multiple comparison post-test.

**Figure 5 genes-15-00616-f005:**
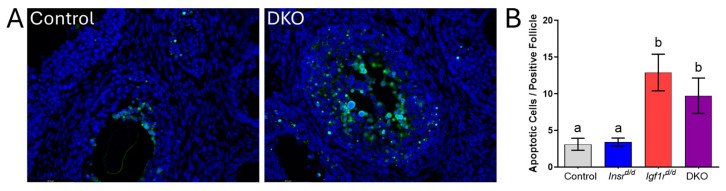
Apoptosis is increased in antral follicles of mice *Igf1r*^d/d^ and DKO. To assess follicle quality, a TUNEL assay was used to assess the frequency of apoptosis in control mice and those lacking *Insr*, Igf1r, or both. (**A**) Representative images of control and DKO mice with TUNEL-positive cells indicated by green fluorescence with DAPI counterstain. (**B**) The quantification of TUNEL-positive cells in antral follicles which contained at least one positive cell (*n* = 12 control, *Insr*^d/d^ and DKO, *n* = 20 *Igf1r*^d/d^). Data are presented as the mean ± SEM. Letters denote means that are significantly different (*p* < 0.05), one-way ANOVA with a Tukey multiple comparison post-test.

## Data Availability

The original contributions presented in the study are included in the article and Appendix A, further inquiries can be directed to the corresponding author (J.A.M.II).

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
