# Peer review of "Normal Ovarian Function in Subfertile Mouse with Amhr2-Cre-Driven Ablation of Insr and Igf1r"

_genes, 2024, doi:10.3390/genes15050616_

Round 1

Reviewer 1 Report

Comments and Suggestions for Authors

The authors used Amhr2-Cre to generate mice with tissue-specific defects of Igf1r and Insr, two receptors involved in insulin signaling in granulosa cells, and analyzed their phenotypes. The induction of genetic defects by Amhr2-Cre is necessary to fill the gap between previous reports of Esr2-Cre/Cyp19-Cre and Pgr-Cre, which differ in their timing in follicle development. In this manuscript, the authors conclude that the induction of target gene defects by Amhr2-Cre occurred in only about 30-40% of the granulosa cells and had no effect on ovarian function. On the other hand, they note that the reduction in litter size in single KOs and DKOs may indicate that functional failure in the endometrium. The usefulness of several Amhr2-Cre lines is also discussed. Although it can be said that this study did not elicit the induction of defects initially expected, it may provide useful information for reproductive function studies using these lines.

The following points should be carefully considered.

1.    Figure 2 graphically shows that Amhr2-Cre induces INSR and IGF1R deletion rates by 27% and 36%, respectively, but it is difficult to read this from the IHC images shown. It is considered necessary to present images at higher magnification. It should also be clarified how the positive and negative cells were evaluated.

2.    Why does the number of cells undergoing apoptosis in the antral follicles increase even though the deletion rate is about 30-40% and does not significantly affect ovarian function? This should be discussed.

3.    It is very curious to see if the morphological differences in the uterus observed in the Pgr-Cre mice will be similarly observed in the mice generated in this study, although the uterus is still under analysis. Information on the uterus, at least a gross morphology and/or histological analysis, should be provided.

L69 Reference [9] and L73 Reference [12] are not properly cited.

Reviewer 2 Report

Comments and Suggestions for Authors

This study investigated the function of insulin receptor (Insr), insulin-like growth factor-1 receptor (Igf1r), and the compensatory action of these two receptors in the secondary and antral follicles of the ovary by utilizing conditional knockout mouse models. The novel aspect of this study is the functional deletion of Insr and Igf1r specifically in the secondary and antral follicles as opposed to other Insr and Igf1r conditional knockouts (driven by Esr2-cre, Cyp19-cre, Pgr-cre, etc.) that have studied different special-temporal gene deletion among reproductive organs and their respective cell types.

This reviewer’s comments are as follows:

1.)     The authors indicate E2, P4, and LH as primary hormones that guide follicle growth and ovulation (lines 43-45). Please include follicle-stimulating hormone (FSH) in this list as FSH is essential to follicle growth, and the authors refer to FSH action throughout the manuscript.

2.)     In figure 1 (lines 197-199) and figure 4 (lines 296-298) authors indicate that student’s t test was used as the statistical test to compare treatment groups. As this study included more than 2 treatment groups, an ANOVA followed by a multiple comparisons test should be used for statistical analysis. Please use ANOVA followed by multiple comparisons for these statistical analyses. Interestingly, the authors do use ANOVA followed by multiple comparisons for the statistical test in the study represented in figure 5 (318-320).

3.)     In the materials and methods section, please include a section describing the statistical analyses used.

4.)     Materials and methods, 2.1 Mice-Please include Esr2-cre as this mouse model was used for figure 3B.

5.)     Please indicate from where mouse lines (Igf1r f/f, Amhr2-cre, Insr f/f, Esr2-cre) were obtained (vendor, collaborator, etc.) If these lines were created in the laboratory, please indicate this.

6.)     Figure 3C – please include a title for the y-axes

7.)     The authors speculate that because ovulation is unaffected in the knockout models, then there must be a defect in implantation and decidualization due to uterine stromal cell deletion of insulin receptor signaling (lines 22-24, 371). Additionally, authors indicate that Amhr2-cre is active in granulosa cells of the ovary and in uterine stromal cells (331-334) and  The Amhr2tm3(cre)Bhr mouse has cre-recombinase activity that can result in gene loss of function within the oviduct (1). Additionally, insulin-like growth factor receptors are present in the oviduct (2). The oviduct is essential for fertilization and embryo development. It is possible that there is a loss of Igf1r in the oviduct, which could affect the viability of embryo. Thus, the subfertility as shown by reduced litter size in figure 1C, could also be a result of embryo viability in addition to or instead of a uterine specific defect. Please comment. 
